# The Role of Histamine in the Pathophysiology of Asthma and the Clinical Efficacy of Antihistamines in Asthma Therapy

**DOI:** 10.3390/ijms20071733

**Published:** 2019-04-08

**Authors:** Kohei Yamauchi, Masahito Ogasawara

**Affiliations:** 1Division of Respirology, Department of Internal Medicine, Takizawa Central Hospital, Takizawa 020-0664, Japan; 2Division of Bioregulatory Pharmacology, Department of Pharmacology, Iwate Medical University, Morioka 028-3694, Japan; ogmasajn0408@gmail.com

**Keywords:** airway obstruction, histamine receptor, histamine receptor 1 antagonist, histamine transporter, OCT-3, Th2 cell, allergen, IgE, innate immunity, type 2 innate lymphoid cells, allergic rhinitis

## Abstract

Mast cells play a critical role in the pathogenesis of allergic asthma. Histamine is a central mediator released from mast cells through allergic reactions. Histamine plays a role in airway obstruction via smooth muscle contraction, bronchial secretion, and airway mucosal edema. However, previous clinical trials of H1 receptor antagonists (H1RAs) as a treatment for asthma were not successful. In recent years, type 2 innate immunity has been demonstrated to be involved in allergic airway inflammation. Allergic asthma is defined by IgE antibody-mediated mast cell degranulation, while group 2 innate lymphoid cells (ILC2) induce eosinophilic inflammation in nonallergic asthma without allergen-specific IgE. Anti-IgE therapy has demonstrated prominent efficacy in the treatment of severe allergic asthmatics sensitized with specific perennial allergens. Furthermore, recent trials of specific cytokine antagonists indicated that these antagonists were effective in only some subtypes of asthma. Accordingly, H1RAs may show significant clinical efficacy for some subtypes of allergic asthma in which histamine is deeply associated with the pathophysiology.

## 1. Histamine Receptors in the Lung

It has been elucidated that four types of histamine receptors such as H1, H2, H3, and H4 exist in the airway and pulmonary tissue [1,2,3,4].

The bronchoconstriction of smooth muscle mediated via H1 receptors is one of the most well-known biological actions of histamine in the respiratory system. It was reported long before that histamine evoked a contraction of human bronchi, and bronchoconstriction was recognized first as one of the biological actions of histamine [5]. While histamine contracts bronchial smooth muscles as strongly as muscarinic M1 receptor agonists, histamine contracts pulmonary peripheral tissue samples more strongly than M1 receptor agonists. This result seemed to suggest the higher sensitivity of peripheral airways to histamine, although it was possible that the contraction of vascular smooth muscles was involved in the contraction of the pulmonary peripheral tissue samples because the sample contained vessels [6].

The response of pulmonary arteries to histamine is biphasic induced by vascular contraction via H1 receptors and vascular dilation via H2 receptors [7]. Histamine induces plasma leakage from postcapillary venules by affecting the bronchial microcirculatory system.

Histamine increases the secretion of mucous glycoprotein from the human airway in vitro. This action is inhibited by the H2 receptor antagonist (H2RA), cimetidine, not H1 receptor antagonists (H1RAs). Histamine also accelerates the chloride ion transport of airway epithelial cells, which is closely associated with water transfer in the airway [8].

## 2. Histamine and Mast Cells in Asthma

Histamine has been a well-known chemical mediator released from mast cells in the immediate allergic reaction for a long time and has been thought to have a critical role in the asthma pathophysiology [9]. Histamine is released into the surface of the airway by inhaled allergens and direct contact with a bronchoscope, and is recovered in bronchoalveolar lavage fluid (BALF). Under the same condition, other chemical mediators are released from mast cells, such as 9α, 11β-prostaglandin F2-α, and tryptase, and are also recovered in BALF. In fact, it was reported that the histamine concentration in the BALF of patients with asthma was significantly higher than that from patients with allergic rhinitis [10]. Furthermore, Tomioka et al. estimated that the number of mast cells in BALF of asthmatic patients was greater than that of control subjects [11] (Figure 1).

Histamine is a representative bioamine that exerts strong and diverse biological actions, and was the first chemical mediator historically found to evoke bronchial smooth muscle contraction in asthma. Dale and Laidlaw first reported the action of bronchoconstriction by histamine [12]. They reported that the bronchoconstriction of guinea pigs induced by venous injection of histamine is similar to that observed by allergen challenge to sensitized guinea pigs with antigen. Therefore, they suggested that histamine was an important chemical mediator that evokes bronchoconstriction in allergen challenge and might be involved in airway obstruction.

Since this study suggested that the airways of guinea pigs were the most sensitive to histamine in terms of undergoing airway smooth muscle contractions, the guinea pigs were used most frequently throughout the twentieth century as an animal model of asthma induced by allergen challenge. In addition, almost all histamine in the skin is localized in mast cells as in the mast cells that exist in the airways. They determined that histamine released from mast cells was involved in the airway response after allergen challenge. Furthermore, since IgE and high-affinity IgE receptors on the surface of mast cells were identified, the concepts of allergen sensitization and the release of chemical mediators such as histamine from mast cells were combined [13]. Based on this historical background, it was clearly recognized that histamine was an important chemical mediator involved in the bronchoconstriction induced by an immediate reaction in bronchial asthma.

## 3. Histamine in the Pathophysiology of Asthma

Curry reported a historical finding concerning asthma pathophysiology, showing histamine induced bronchoconstriction in asthmatics by injection or inhalation at a low dose of histamine that had no effect in normal subjects [14]. Based on these results, the concept of airway hyperresponsiveness to histamine as a physical characteristic of asthmatics was proposed. Afterward, since the airways of asthmatics were hyperresponsive to many airway smooth-muscle-contracting agents, it has been recognized as nonspecific airway hyperresponsiveness showing airway abnormality [15].

Terfenadine, a strong H1RA, induced bronchodilation in asthmatics the same as β2-stimulant. This result suggested that the continuous release of histamine from mast cells in the lung of asthmatics was causing contractive tension of the airway smooth muscle. These lines of evidence suggested that histamine was deeply involved in the asthma pathophysiology. Furthermore, terfenadine reversed completely the decrease in the forced expiratory volume in one second (FEV1) as an index of airway obstruction induced by histamine. On the other hand, terfenadine reversed partially the decrease of FEV1 induced by allergen inhalation (Figure 2) [16].

It was reported that the histamine concentration in the BALF of patients with asthma was significantly higher than that of normal subjects as shown Figure 1 [12]. Furthermore, mast cells and basophils were increased in the BALF of asthmatics during exacerbations [13]. The mRNA level of l-histidine decarboxylase, a histamine synthase, was markedly elevated in the pulmonary tissue of patients with asthmatic death [17].

Histamine *N*-methyl transferase (HMT) is the principal enzyme that metabolizes histamine in the airway [18]. HMT activity was measured in human trachea and bronchi. In addition, the contractile response of isolated human bronchi to histamine was potentiated in the presence of an HMT inhibitor, SKF 91488. These results suggest that HMT plays an important role in degrading histamine and in regulating the airway response to histamine [19] (Figure 3). In addition, it was reported that polymorphisms of H1R and HMT gene in the patients with allergic asthma were significantly different from those of nonallergic asthma, suggesting that polymorphisms of H1R and HMT gene were involved in the pathogenesis of allergic asthma [20].

## 4. Histamine Transport in the Pathophysiology of Asthma

Histamine is synthesized and stored in the vesicles of mast cells and basophils [21]. Upon immunological stimulation of mast cells and basophils, histamine is released from storage vesicles into the extracellular space activating G-protein-coupled receptors H1, H2, H3, and H4 [21]. However, to terminate the effects of histamine via histamine receptors on targets cells such as bronchial smooth muscle cells, the histamine concentration in the extracellular space should be regulated by the degradation of histamine.

The degradation enzyme, HMT, is critical in metabolizing histamine into inactive forms of the metabolite and was documented to be significant for the relationship between airway responsiveness and HMT activities when using the HMT inhibitor, SFK91488, in animal models [16,22]. Biochemical analysis suggested that the HMT enzyme is primarily localized in cytoplasmic space [23], while histamine is unable to easily enter the intracellular space because at physiological pH, histamine exists as an organic cation. Therefore, transport machineries of histamine are required to enter the intracellular space and to obtain access to the HMT enzyme.

We hypothesized the two potential mechanisms: (1) the HMT enzyme translocates to the plasma membrane upon some stimulation and possibly achieves access to histamine (membrane translocation hypothesis) and (2) other molecules assist the transport of histamine into cells (transporter hypothesis). In terms of the first hypothesis (membrane translocation hypothesis), we clarified the translocation of HMT to the plasma membrane upon stimulation of adrenergic receptor with isoproterenol [23]. In regard to the second hypothesis (transporter hypothesis), supportive evidence was reported for comparatively high-capacity transporters (solute carrier cation transporters 22As; SLC22As) in the reuptake of endogenous substrates, such as norepinephrine, dopamine, serotonin, and histamine [24]. 

Among these transporters, organic cation transporter (OCT)-2 (SLC22A2) and -3 (SLC22A3) revealed the ability to transport histamine into cells via a potential-sensitive mode. OCT-2 was exclusively expressed in the kidney, while OCT-3 was ubiquitously expressed [24]. In order to clarify the molecular mechanisms of how the histamine transport system is associated with HMT inactivation, we prepared three types of cells stably expressing HMT alone, plasma membrane targeting form of HMT due to *N*-terminal myristylation, and HMT plus OCT-2 for in vivo/in vitro assays followed by measuring the concentration of extracellular as well as intracellular histamine. Cells expressing HMT with OCT-2 indicated significantly increased intracellular histamine contents compared with cells expressing HMT with myristylation modification [23], suggesting that the transporter is critically required for histamine metabolism.

Based on previous studies on the distribution and capability of histamine transport, organic cation transporter-3 (OCT-3) is presumed to be the most important transporter in allergic asthma, although the functional characterizations of all transporters identified to date have not been fully completed in regard to histamine transporter activities. In general, both membrane drug transporters and metabolic enzymes are involved in the clearance of drugs [24,25]. Likewise, both OCT-3 and metabolic enzyme HMT are intimately interconnected and operate in the reduction of the extracellular histamine concentration. Therefore, alterations in the histamine transport activities in the respiratory tissues would critically influence the pharmacokinetics of histamine. Oct3/Slc22a3 (mouse homologue of human OCT-3)-deficient mice were generated [26] and exhibited increased accumulation of histamine in the brain as well as in the peripheral plasma [27].

As far as we know to date, there are 11 studies on functional and phenotypic alterations suspected to be related to human OCT-3 polymorphisms [28,29,30,31,32,33,34,35,36,37]. Yamauchi et al. reported that four major genetic polymorphisms (S116A, R120R, I140T, and A411A) of the *OCT-3* gene including synonymous alterations were investigated concerning the relationship between the allele frequency and asthma severities, and were classified into two groups (mild and moderate/severe) on the basis of the Japanese asthma severity guidelines. Patients with moderate and/or severe symptoms exhibited significantly lower frequency of the C allele compared to those showing mild symptoms, suggesting that a genetic alteration of the *OCT-3* gene may have relevance to the asthma severity [28].

The functional analyses of human OCT-3 genetic polymorphisms were performed in regard to histamine transporter activities employing heterologous expression of OCT-3 with five genetic polymorphisms (T44M, A116S, T400I, A439V, and G475S) followed by measurement of the histamine and MPP+ transport activities. Of the OCT-3 mutants analyzed, four (A116S, T400I, A439V, and G475S) exhibited ~50–75% of the histamine transporter activity as compared to controls [29]. Of note, concerning the T400I mutant, which was investigated in an association study of asthma patients, there was no clear significance in the relationship between the allele frequency of the *OCT-3* gene and asthma severities.

On the other hand, OCT-3 with T400I mutation could have the potential to increase the concentration of histamine in the extracellular space, possibly resulting in higher stimulation of H1R, which is associated with the induction of bronchial smooth muscle contraction, epithelial barrier dysfunction, and increased secretion of mucus. Difference between the functional analysis results and the clinical genetic outcomes may be explained by two possibilities, based on previous studies [24,38].

One of the possibilities is that, because epinephrine is also a good substrate for OCT-3, the decreased affinity of epinephrine for OCT-3 compared with that of the control OCT-3 would increase the concentration of epinephrine induced in the extracellular space, counteracting the H1R stimulation [24]. The other possibility is that, because a higher concentration of histamine was induced in the extracellular space due to the decreased histamine transporter activities of OCT-3, H2R could have been more stimulated in addition to H1R stimulation.

Recently, a novel function of H2R in inhibiting proinflammatory responses in the lung was elucidated using H2R genetic deficiency and the administration of an H2RA, famotidine [38]. H2RA treatment or H2R-deficient animals showed increased numbers of CD1d+ dendritic cells and iNKT cells, resulting in inflammatory cell recruitment, and Th2 cytokine productions and secretion such as IL-4 and IL-5, while dimaprit, a selective H2R agonist, inhibited the lung iNKT cell responses [38]. Furthermore, the OCT-3 functions are more complicated in terms of the bidirectional transport system of OCT-3; inward histamine transport system in bronchial epithelial cells expressing HMT is critical for the clearance and inactivation of histamine. On the other hand, in the outward histamine transport system in professional histamine-producing cells such as mast cells and basophils expressing histamine-producing enzyme, histidine decarboxylase (HDC) regulates the intracellular contents of histamine through homeostatic mechanisms [39] (Figure 4A,B).

The results of the genetic studies and functional analysis suggested that genetic variants of OCT-3 transporter have critical effects on the clearance of extracellular histamine, monoamines such as epinephrine, and homeostatic regulation of histamine production contributing to a variety of outcomes. Studies on the functions and polymorphisms of OCT-3 could increase our understanding of important factors in the pathophysiology of asthma.

## 5. H1RA in Asthma Therapy

In the 1940s, Herxheimer demonstrated significant airway dilation in asthmatics by administering chlorpheniramine, suggesting its clinical usefulness [40]. Subsequently, it was confirmed that other H1RAs such as clemastine induced airway dilation by administration per os or by inhalation. However, since the earlier-generation H1RAs contained inhibitory activities on smooth muscle contraction through muscarinic, adrenergic, and serotonergic receptors, the action of airway dilation was not determined to be caused by antihistaminic action via H1 receptor alone.

In the 1980s, H1RAs with high selectivity for H1 receptor and strong biological activity but weaker effects on CNS were developed. Since these new-generation H1RAs induced airway dilation, the role of the H1 receptor in the smooth muscle of the airway was determined. H1RAs are recommended in the therapy for allergic rhinitis and urticaria, which are thought to be caused by immediate reactions [41,42,43]. However, H1RAs have been thought to be supplementary therapeutics in asthma guidelines and their role in asthma therapy is not extensive. The primary drug of asthma therapy is the inhaled corticosteroid (ICS). In addition, long-acting β2 agonist (LABA), theophylline, short-acting β2 agonist (SABA), and leukotriene receptor antagonist (LTRA) are recommended for asthma in asthma guidelines (Table 1) [44].

Although mast cells and histamine were thought to be deeply associated with the asthma pathogenesis and pathophysiology as described above, H1RAs were generally not recommended for asthma therapy. The following reports may explain one of the reasons for the ineffectiveness of H1RAs in asthma therapy. Terfenadine reversed completely the decrease in FEV1 as an index of airway obstruction induced by histamine. On the other hand, terfenadine reversed partially the decrease of FEV1 induced by allergen inhalation (Figure 2). There was a report comparing the effectiveness between LTRAs and H1RAs on airway obstruction in immediate and late airway response after allergen inhalation. That report revealed that LTRAs were superior to H1RAs in inhibiting airway obstruction induced by allergen challenge (Figure 5) [45]. As a consequence, LTRAs have a principal role in the therapeutics for asthma in the asthma guidelines, and H1RAs are thought to be supplementary drugs in asthma therapy.

## 6. The Immunological Roles of Histamine in Allergic Reactions

To date, histamine has been thought to play critical roles in airway obstruction through airway smooth muscle contraction, acceleration of secretion from airway submucosal glands, and airway submucosal edema. In addition, histamine has immunological roles in the actions of dendritic cells, B cells, and Th1 and Th2 lymphocytes through H1 and H2 receptors on their cell surface. The histamine receptors on the cell surface were identified by staining with fluorescein-labelled histamine; histamine bound significantly more strongly to Th1 lymphocytes than to Th2 lymphocytes [46].

The H1R-specific antagonist tripelennamine inhibited histamine binding in Th1 lymphocytes but not in Th2 lymphocytes. These results indicated that H1R was expressed predominantly on Th1 lymphocytes. Neither the H2RA ranitidine nor the H3R antagonist (H3RA) clobenpropit influenced the histamine binding to Th1 lymphocytes. IL-3 significantly increased histamine receptor expression on Th1 lymphocytes but not on Th2 lymphocytes [46]. The predominant expression of H1R on Th1 lymphocytes and H2R on Th2 lymphocytes is evaluated by specific antibodies against the H1R and H2R (Figure 6).

The predominance of Th1 lymphocytes has been reported in PPD-specific T-cell responses [47], and T-cell proliferation was augmented by histamine in PPD-stimulated cultures. In contrast, allergen-stimulated cells from individuals allergic to house dust mites showed mostly Th2 lymphocytes [48]. In addition, histamine enhanced IFN-γ secretion from Th1 lymphocytes, whereas histamine inhibited secretion of Th2 cytokines (IL-4, IL-13) from Th2 lymphocytes.

It was reported that H1R knockout mice showed higher OVA-specific IgE compared to wild-type mice. In contrast, H2R knockout mice demonstrated lower serum levels of OVA-specific IgE compared to wild-type mice. These results indicate that H1R plays an important role in the suppression of the humoral immune response including IgE production.

Ohtsu et al. reported histamine-deficient mice with disrupted *HDC* gene [49]. These HDC knockout mice were viable and fertile, but the numbers of mast cells were decreased compared with the wild mice, while the remaining mast cells showed an altered morphology and reduced granular content. 

The role of endogenous histamine in eosinophilic recruitment and hyperresponsiveness was examined in a murine asthma model using HDC knockout mice [50]. The histamine contents in the bronchial tissue in HDC knockout mice were markedly decreased compared with wild mice. Inhalation challenge with OVA in OVA-sensitized wild mice caused eosinophil accumulation in the airways as well as airway hyperresponsiveness. On the other hand, the eosinophil recruitment to lung was significantly reduced in HDC knockout mice, suggesting that endogenous histamine is involved in the accumulation of eosinophils into the airways in allergic reaction. Since histamine is reported to have eosinophil chemotactic activity via H4R [51], reduced eosinophils in HDC knockout mice could be explained through the activity via H4R.

Systemic anaphylaxis is occasionally fatal and is thought to be caused by systemically circulated histamine released from mast cells activated with allergens. The reaction starts from a few minutes to 30 min after the allergen challenge, and it leads to an increase in vascular permeability, contraction of smooth muscle, and an increase in mucin secretion; the final results are hypotension, tachycardia, and a decrease in peripheral blood resistance, sometimes leading to death. To evaluate the role of histamine in systemic anaphylaxis, the responses in body temperature, blood pressure, and respiratory function in HDC knockout mice were compared with wild mice [52]. The respiratory frequency dropped and the body temperature was decreased only in HDC knockout mice. Therefore, respiratory frequency and body temperature were thought to be controlled by histamine.

Bronchial asthma is characterized by goblet cell hyperplasia and mucus overproduction and goblet cell hyperplasia with mucus hypersecretion has been reported to be associated with the development of airway hyperresponsiveness and an increase in the severity and mortality in bronchial asthma. Histamine is known to strongly stimulate goblet cell secretion. Bryce et al. reported that goblet cell hyperplasia in response to allergens was reduced in H1R knockout mice, suggesting that histamine may be involved in the immunomodulation of the airway allergic reaction including Th1 and Th2 cytokine production and goblet cell hypersecretion. To examine the role of histamine in goblet cell hyperplasia in asthma, the number of goblet cells, *Gob-5* gene expression, and cytokines in BALF in HDC knockout mice were compared with those in wild mice [53]. Unexpectedly, PAS staining demonstrated an increase in the number of goblet cells in the epithelium in large and small airways of the HDC knockout mice sensitized with OVA compared to those of wild mice sensitized with OVA. In addition, the level of Gob-5 mRNA in the HDC knockout mice exposed to OVA was significantly higher than that in the wild-type mice under the same conditions.

Previous reports have suggested that the expression of the *Gob-5* gene, which corresponds to hCLCA1 in human beings, could be one of the first steps in mucus cell metaplasia and hyperplasia by inducing mucin gene expression [54]. The concentration of TNF-α in BALF of the HDC knockout mice exposed to OVA was significantly higher than that of the wild-type mice under the same conditions, while, in contrast, the concentration of IL-4 in BALF of the HDC knockout mice exposed to OVA was significantly lower than that of the wild mice under the same conditions. Antigen challenge of sensitized mice or human asthmatic subjects results in increased TNF-α expression in BALF, peripheral blood, and tissue biopsy specimens [55]. TNF-α production is partly regulated by histamine via H2R in macrophages and dendritic cells. Busse et al. have recently demonstrated that chronic exposure of TNF-α to the airway induced goblet cell hyperplasia [56]. Taking these findings into consideration, the depletion of histamine in the HDC knockout mice exposed to OVA might have induced the upregulation of TNF-α production, resulting in the increase of goblet cells in the airways with allergic inflammation.

In addition, as we described above, the increase in H1R expression on Th1 lymphocytes by IL-3 may account for negative-feedback regulation due to an antiallergic effect of Th1 lymphocytes. Combined with the results of the enhanced goblet cell hyperplasia in the murine asthma model with HDC knockout mice, it is suggested that histamine secreted from effector cells potently influences the Th1 and Th2 responses as a regulatory loop in inflammatory reactions. Consequently, the inhibition of histamine actions does not always attenuate asthmatic reactions.

## 7. A New Insight on the Immunological Pathway in Asthma

Asthma is an inflammatory disorder of the conducting airways that has a strong association with allergic sensitization. The disease is characterized by a polarized Th-2 (T-helper-2)-type T-cell response and the levels of IL-4, IL-5, and IL-13 are upregulated in the airways of patients with bronchial asthma [57]. After an exposure to allergens, specific IgE on the high-affinity IgE receptors of mast cells conjugates with allergens, IgE and allergen complex activate mast cells through IgE receptors and release chemical mediators including histamine, leukotrienes, prostaglandins, and so forth. This immunological event causes an immediate reaction in the airways of asthmatics.

However, over the past decade, the understanding of asthma pathogenesis has made a significant shift from a Th2 cell-dependent, IgE-mediated disease to a more complicated heterogeneous disease. Recent studies clearly show that not only Th2 cytokines but also other T-cell-related cytokines such as IL-17A and IL-22 as well as epithelial cell cytokines such as IL-25, IL-33, and thymic stromal lymphopoietin (TSLP) are involved in the pathogenesis of asthma. Recently, type 2 innate lymphoid cells (ILC2) were found to represent a critical innate source of type 2 cytokines [58,59,60]. In the classical immunological pathway, the role of mast cells is critical in the immediate reaction.

On the other hand, recent advances in the immunoreactions of asthma revealed that in the innate immune response, epithelial cells release innate immune molecules such as IL-33, TSLA, and so forth in response to foreign stimulants such as microorganisms, dust, smoke, and so forth. Consequently, TSLP or IL-33 activates ICL2 to release various Th2 type cytokines such as IL-5 and IL-13, inducing eosinophilic inflammation in the airways. This pathway of innate immunity explains well the pathogenesis of asthmatic acute exacerbations caused by infection or inhalation of dusty air. However, mast cell activation is not involved in the pathway of the innate immunity of asthma pathogenesis.

## 8. A New Aspect of H1RA and H4RA in Asthma Therapy

For three decades, the primary therapies for asthma included ICS and LABA, LAMA, theophylline, and LTRA, combined with ICS depending on the severity [61,62]. As described above, the clinical trials of H1RA for asthma have not been successful and recent advances in immunology revealed IgE-independent immunological pathways, suggesting a critical immune pathway without mast cells in the asthma pathogenesis. These lines of evidence suggest that ICS may be enough for asthma therapy and that the significance of mast cells in asthma is small.

However, Omalizmab, an anti-IgE antibody, demonstrated excellent clinical effects on severe allergic asthmatics [63]. This clinical effect reminded us that mast cells are still playing critical roles in the asthma pathophysiology. Since asthma is a disease with heterogenous immunological pathways including IgE-independent pathways, H1RAs would not be effective in the general population of asthmatics. In other words, H1RAs can be expected to be clinically effective in some subtypes of asthma.

Bronchial asthma and allergic rhinitis (AR) are thought to share a common pathogenesis. More than 60 percent of patients with adult asthma showed symptoms of AR and were diagnosed with AR. In addition, asthma and AR symptoms often co-occur in patients with asthma and AR (Table 2) [64]. H1RAs are recommended for AR in clinical guidelines. In this regard, asthmatics with AR may represent a promising subtype in which H1RAs may be very effective on both asthmatic and AR symptoms (Figure 7). Allergic asthmatics are thought to represent more than 70% of total asthmatics. Therefore, asthmatics with AR are a large subtype.

In addition, therapy with H1RAs could be promising for other subtypes of asthmatics with allergic characteristics. The easy detection of the histamine content in serum would help in identifying promising subtypes of asthmatics suitable for therapy with H1RAs. 

H4 receptor has been recognized to play a critical role in the inflammatory response involved in the pathogenesis of asthma. Stimulation of H4R can also enhance the migration of eosinophils and the recruitment of mast cells leading to the amplification of immune responses and chronic inflammation [4,51,65,66]. Similarly, H4R is involved in T-cell differentiation and dendritic cell activation, and its immunomodulatory function [67]. H4R antagonists (H4RAs) have previously been shown to have anti-inflammatory activity in a murine asthma model [68]. In addition, Cowden et al. demonstrated that the H4RA JNJ 7777120 attenuated airway hyperreactivity and ameliorated the asthmatic airway remodeling including goblet cell hyperplasia and airway collagen deposition in the murine asthma model [69]. Accordingly, H4RAs were expected to be clinically effective for asthma. Thurmond et al. reported that a selective H4RA, JNJ-39758979, exhibited good preclinical and phase 1 safety in healthy volunteers with evidence of a pharmacodynamics effect in humans [70]. Kollmeier et al. reported a phase 2a study of H4RA, toreforant, in which its clinical efficacy on eosinophilic asthma was not evaluated [71]. In the near future, new trials of another selective H4RA in other subtypes of asthma are expected.

## Figures and Tables

**Figure 1 ijms-20-01733-f001:**
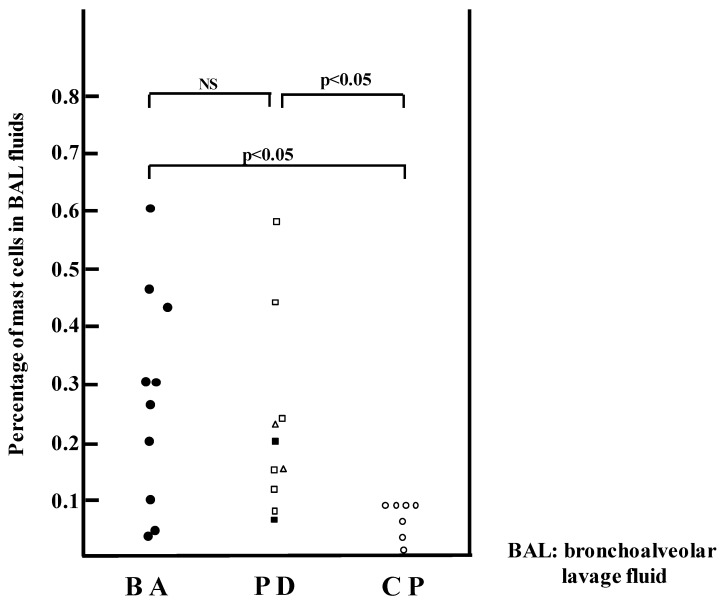
Percentage of mast cells in BAL fluids in asthmatics (BA: ●), patients with other pulmonary diseases (PD): interstitial pneumonia (□), sarcoidosis (△), and chronic obstructive pulmonary diseases (■), and control patients (CP: ○). (Bars indicate SEM). The figure was modified from reference [11] with permission: Tomioka et al., Mast cells in bronchoalveolar lumen of patients with bronchial asthma. Am. Rev. Respir. Dis. 1984, *129*, 1000–1005.

**Figure 2 ijms-20-01733-f002:**
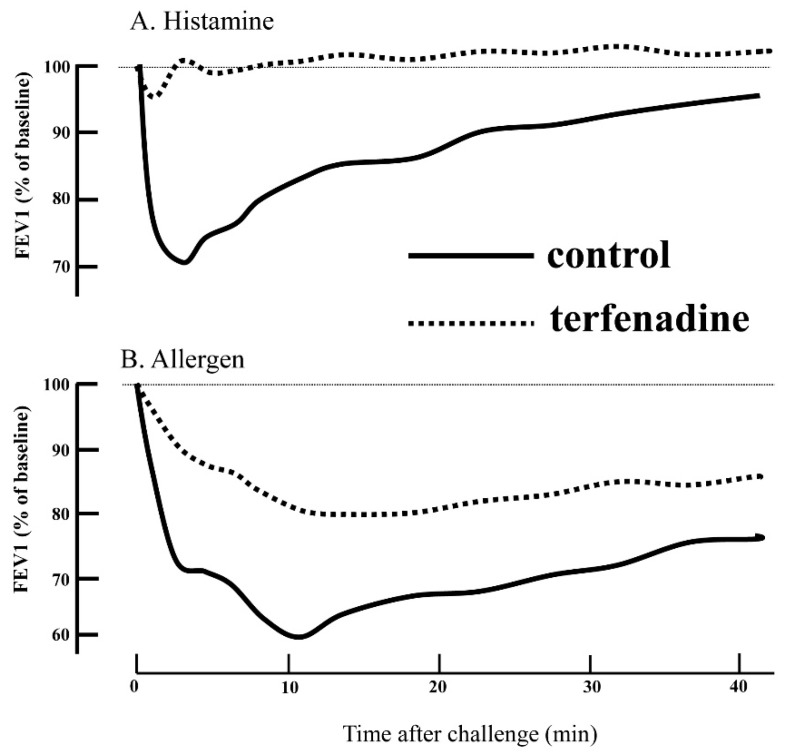
The effects of terfenadine on bronchoconstriction induced by (**A**) Histamine and (**B**) Allergen. Terfenadine (dotted curve) significantly attenuated bronchoconstriction induced by allergic challenge. (Solid curves indicate pretreatment with terfenadine. Dotted curves indicate pretreatment with placebo. The figure was modified from reference [16] with permission: Rafferty et al., The contribution of histamine to immediate bronchoconstriction provoked by inhaled allergen and adenosine 5’ monophosphate in atopic asthma. Am. Rev. Respir. Dis. 1987, *13,* 369–373.

**Figure 3 ijms-20-01733-f003:**
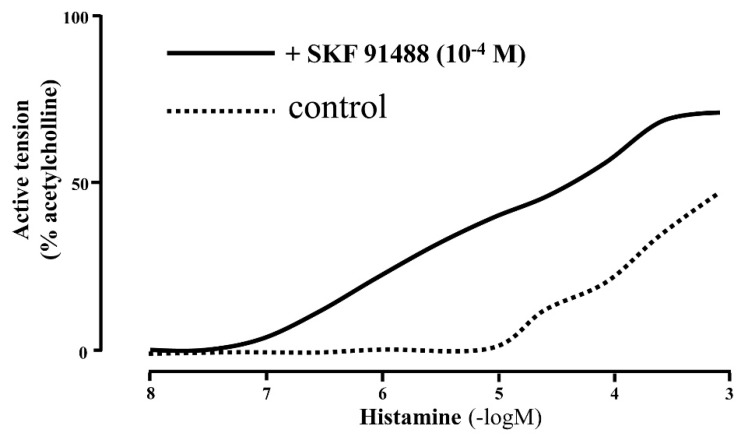
Effect of SKF 91488 and aminoguanidine on contractile response of human bronchi to histamine. Dose–response curves to histamine in SKF 91488 (10^−4^ M) and control are analyzed. Data are expressed as percentage of response to acetylcholine (10^−3^ M). Histamine N-methyltransferase (HMT) regulates contraction of airway smooth muscle by histamine. HMT in epithelium degrades histamine actively. The figure was modified from reference [19] with permission: Yamauchi et al., Structure and function of human histamine *N*-methyltransferase: critical enzyme in histamine metabolism in airway. Am. J. Physiol. 1994, *267*, L342–L349.

**Figure 4 ijms-20-01733-f004:**
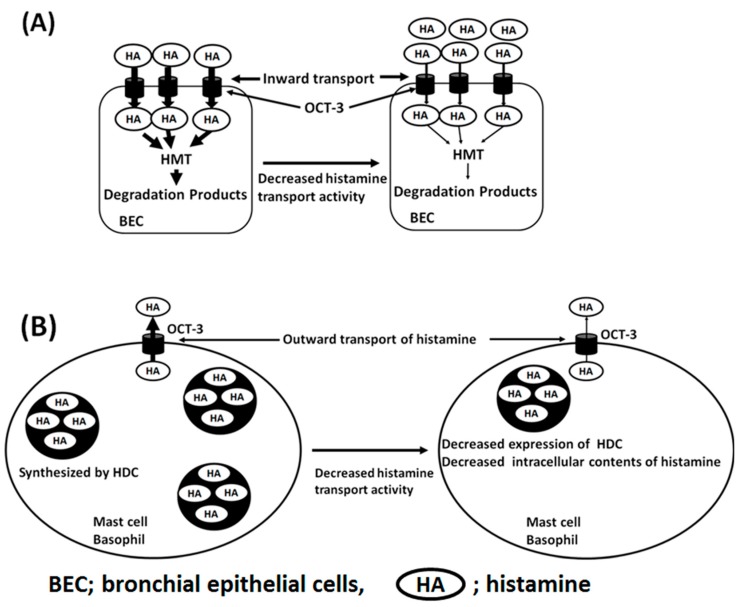
Scheme of histamine transport system through OCT-3. (**A**) Inward transport of histamine in bronchial epithelial cells. Genetic polymorphisms of OCT-3 affect the inward transport activity of histamine, resulting in the increased amounts of histamine in extracellular space and possibly confer stronger stimulation on H1R and induce contraction of bronchial smooth muscle cells. (**B**) Outward transport system of histamine through OCT-3 in professional histamine-producing cells (mast cells and basophils). Genetic polymorphisms of OCT-3 affect the outward transport activity of histamine, resulting in the decreased amounts of histamine contents in the vesicles through homeostatic mechanisms.

**Figure 5 ijms-20-01733-f005:**
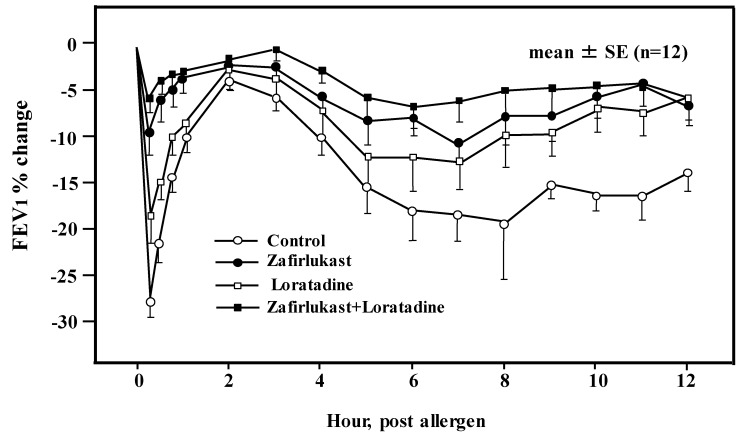
Time course of allergen-induced early and late airway reaction after different treatment strategies in the subjects. Pulmonary function, expressed as percent of postdiluent FEV 1, was followed hourly for 12 h. The figure was modified from reference [45] with permission: Roquet et al., Combined antagonism of leukotrienes and histamine produces predominant inhibition of allergen-induced early and late phase airway obstruction in asthmatics. Am. J. Respir. Crit. Care Med. 1997, *155*, 1856–1863.

**Figure 6 ijms-20-01733-f006:**
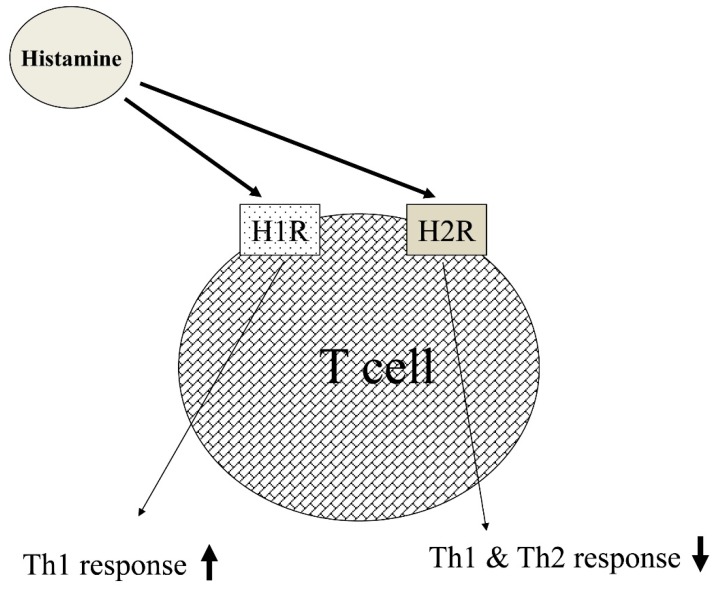
Regulation of Th1 and Th2 responses by histamine via H1R and H2R.

**Figure 7 ijms-20-01733-f007:**
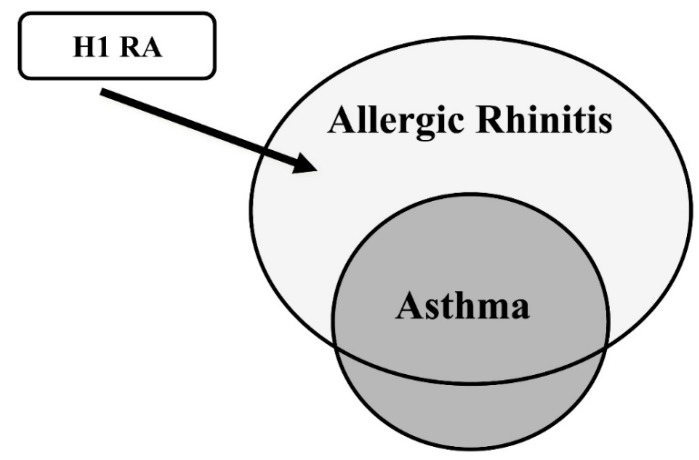
The therapeutic use of H1RAs to the comorbidity of asthma and allergic rhinitis. More than 60% of asthmatics represented the comorbidity with allergic rhinitis.

**Table 1 ijms-20-01733-t001:** Standard treatment steps for adult asthma.

	Step 1	Step 2	Step 3	Step 4	Step 5
Controllers		Low dose ICS	Low dose ICS/LABA	Med/high ICS/LABA	Add-on Tiotropium, Anti-IgE, Anti-IL-5
Other controllers options	ICS (low dose)	LTRA Low dose Theophylline	Med/high dose ICS; low dose ICS + LTRA (or + Theophylline)	Add Tiotropium Med/high dose ICS; low dose ICS + LTRA (or + Theophylline)	Add low dose OCS
Relievers	SABA	SABA	SABA or low dose ICS/formoterol		

Controllers: drugs for long-term management; relievers: drugs for exacerbation. ICS: inhaled corticosteroid. LTRA: leukotriene receptor antagonist. LABA: long-acting b2 agonist. SABA: short-acting b2 agonist. OCS: Oral corticosteroid.

**Table 2 ijms-20-01733-t002:** Comorbidity of Asthma with Allergic Rhinitis.

	Number	Yes	No	n.d.
Adult asthma	2781	1693 (60.8%) *	1044 (37.5%)	44 (1.6%)
Child asthma	3283	2238 (68.2%) **	1035 (31.5%)	10 (0.3%)
Allergic rhinitis (AR)	3945	1935(49.0%)	2010 (51.0%)	–

* Adult asthma vs AR *p* < 0.001; ** Child asthma vs. AR *p* < 0.001 (*x*^2^ analysis). Reference [64], Yamauchi et al. Allergol. Int. 2009, *58*, 55–61.

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
