# Peer review of "The Role of Histamine in the Pathophysiology of Asthma and the Clinical Efficacy of Antihistamines in Asthma Therapy"

_ijms, 2019, doi:10.3390/ijms20071733_

Reviewer 1 Report

The authors review the role of histamine in asthma, supporting anti-H1 therapeutic approaches for certain forms of asthma. The topic is of interest but I have several concerns with the current manuscript in its present form. Firstly, it provides more of a historical overview of the role of histamine in asthma rather than more recent advances. The recommended asthma guidelines include the use of theophylline. This is not standard, especially for milder forms of asthma. The authors do not cite any source for this information and it should be amended according to the latest international guidelines (e.g. from reputable societies such as EAACI, AAAAI etc.). The authors seem to support a role for anti-H1 therapy in certain forms of asthma but it is not clear what they mean here. For example, they state: “Since asthma is a disease with heterogenous immunological pathways including IgE-independent pathways, H1RA would not be effective in the general population of asthmatics. In other words, H1RA can be expected to be clinically effective in some subtypes of asthma.” This reasoning is not at all clear, also because mast cells (and basophils) can be activated by IgE-independent means too. Do they mean that anti-H1 would only work in IgE-dependent forms of asthma? If so, the fact remains that anti-histamines have little clinical effect in allergen-(IgE-mediated) induced asthma by themselves, probably due to the redundancy of histamine in causing bronchoconstriction because of the actions of LTC4 and other eicosanoids. The authors actually mention an earlier study that demonstrates a combinatorial approach of anti-H1 + LTRA (Fig2), but this does not feature in their overall message. It would also help if the authors could show which clinical trials using anti-H1 benefited different types of asthma since, as it is, the discussion is rather speculative.

Minor points:

Please amend the title: “….Anti-histamines in Asthma Therapy”

There are numerous minor grammatical mistakes. For example, in the introduction: “…mediated via H1 receptor is one….” should be either corrected to:  “…mediated via H1 receptors is one….” or “…mediated via the H1 receptor is one….”. There are numerous errors such as this throughout the manuscript.

Some paragraphs are too long (e.g. on page 5)

Some of the figures have been taken from previous publications. Although the authors state that these have been modified, are they certain that they do not need additional publisher permission?

Author Response

Response to the reviewer’s comments

We appreciate the careful reviewing of the manuscript (ID ijms-449820).

We responded to the reviewer’s comments point-by-point as follows.

We underlined the changed or inserted sentences in the original manuscript.

Reviewer 1

Comments and Suggestions for Authors

Comments: 1) The authors review the role of histamine in asthma, supporting anti-H1 therapeutic approaches for certain forms of asthma. The topic is of interest but I have several concerns with the current manuscript in its present form. Firstly, it provides more of a historical overview of the role of histamine in asthma rather than more recent advances. The recommended asthma guidelines include the use of theophylline. This is not standard, especially for milder forms of asthma. The authors do not cite any source for this information and it should be amended according to the latest international guidelines (e.g. from reputable societies such as EAACI, AAAAI etc.).

Response: According to the reviewer’s suggestion, we presented the table of standard treatment steps for asthma recommended in Global Strategy for Asthma Management and Prevention (2018 update).

Comments: 2 ) The authors seem to support a role for anti-H1 therapy in certain forms of asthma but it is not clear what they mean here. For example, they state: “Since asthma is a disease with heterogenous immunological pathways including IgE-independent pathways, H1RA would not be effective in the general population of asthmatics. In other words, H1RA can be expected to be clinically effective in some subtypes of asthma.” This reasoning is not at all clear, also because mast cells (and basophils) can be activated by IgE-independent means too. Do they mean that anti-H1 would only work in IgE-dependent forms of asthma? If so, the fact remains that anti-histamines have little clinical effect in allergen-(IgE-mediated) induced asthma by themselves, probably due to the redundancy of histamine in causing bronchoconstriction because of the actions of LTC4 and other eicosanoids. The authors actually mention an earlier study that demonstrates a combinatorial approach of anti-H1 + LTRA (Fig2), but this does not feature in their overall message. It would also help if the authors could show which clinical trials using anti-H1 benefited different types of asthma since, as it is, the discussion is rather speculative.

Response: As the reviewer pointed out, in what subtype or endotype of asthmatics H1-RA is effective is not clear. In addition, we agree that mast cells are activated by Th2 cytokines and mediators in IgE-independent pathway, however we think the major pathway of the histamine release from mast cells is the high affinity IgE receptor triggering by allergen-IgE complex. High concentration of histamine in BALF was evaluated in the patients with allergic asthma. Although a various mediators are involved in the pathophysiology of asthma as the reviewer suggested, we think histamine play a relatively important role in some subtypes of allergic asthma. However, a new drug trial of H1RA on the subtype of asthma is difficult in the current global circumstances. If subtypes or endotypes of asthma can be clearly evaluated and their definition can be easily determined clinically by some biomarkers, the new drug trial of new H1RA on the subtype of asthma will be considered. Concerning the subtype, asthma with allergic rhinitis is thought to be allergic asthma as described in the manuscript, and this subtype is easily recognizable. In addition, H1RA is recommended for the treatment of allergic rhinitis in the guideline. LTR was more effective in the asthmatics with allergic rhinitis compared with those without allergic rhinitis in combination with ICS therapy. (Price D.B.,et al.Allergy Clin Immunol Int.2003;Suppl1;29). In this regard, I suggested a possible therapeutic effect of HIRA on asthma with allergic rhinitis (Brożek JL, Bousquet J, Agache I, Agarwal A, et al. Allergic Rhinitis and its Impact on Asthma (ARIA) guidelines-2016 revision. J Allergy Clin Immunol. 2017;140:950-958.). As the reviewer pointed out, the description of HIRA on asthma with allergic rhinitis was thought to be speculative. I added the description of H4R in pathophysiology of asthma and clinical significance of H4RA on asthma (line 475- line493).

Minor points:

1)      Comments :Please amend the title: “….Anti-histamines in Asthma Therapy”

Response:  We corrected.

2)      Comments : There are numerous minor grammatical mistakes. For example, in the introduction: “…mediated via H1 receptor is one….” should be either corrected to:  “…mediated via H1 receptors is one….” or “…mediated via the H1 receptor is one….”. There are numerous errors such as this throughout the manuscript.

 Response: We checked and changed. (line 36, 49, 50,140, 143, 145, 146, 148, 149, 153, 156, 157, 158, 161, 163, 165, 172, 175, 179, 181, 183, 185, 189-192, 195, 199-200,202-206, 208, 210-211, 213-216, 218-224, 226-233, 227-234, 235-238, 243, 245-246, 249, 251, 252, 254-257, 298, 301,304, 309, 313, 317, 319-322, 325-329, 332, 333, 336-338, 341, 345, 349-351, 359, 363, 366, 369, 371, 374, 375, 377, 381, 383, 398, 403, 405-408 )

3)      Comments : Some paragraphs are too long (e.g. on page 5)

Response: We corrected.  

4)      Comments: Some of the figures have been taken from previous publications. Although the authors state that these have been modified, are they certain that they do not need additional publisher permission?

Response: We modified them to more symbolic figures. If necessary, we will take the permission.

Reviewer 2 Report

This is a reasonable review. However, there is much poorly integrated text in the review with too many single-sentence paragraphs. There are a number of reference links missing. There is limited coverage of both H3 and H4Rs in asthma. For example, there are more recent studies since 2002  which show eosinophil chemotactic activity and other relevant literature to the review topic pertinent to the role the H4R.

Author Response

Reviewer 2

We appreciate the careful reviewing of the manuscript (ID ijms-449820).

We responded to the reviewer’s comments point-by-point as follows.

We underlined the changed or inserted sentences in the original manuscript.

Comments and Suggestions for Authors

1)      Comments: This is a reasonable review. However, there is much poorly integrated text in the review with too many single-sentence paragraphs.

Response: We reconsidered sentences and rewrote the manuscript.

2)      Comments: There are a number of reference links missing.

Response:  We checked and corrected.

3)      Comments: There is limited coverage of both H3 and H4Rs in asthma. For example, there are more recent studies since 2002 which show eosinophil chemotactic activity and other relevant literature to the review topic pertinent to the role the H4R.

Response: I added the description concerning the role of H4R in pathophysiology of asthma and reviewed the recent clinical trial of H4RA on asthma in last session (line 475- line493). 

Round  2

Reviewer 1 Report

The manuscript has improved but the title is still wrong (anti-histamines not anti-histamine). The authors and editors should check whether showing even adapted and cited figures previously published is ok.

Author Response

Reviewer 1

Comments and Suggestions for Authors

1)      Comment: The manuscript has improved but the title is still wrong (anti-histamines not anti-histamine)

Response: We corrected the title.

2)      Comment: The authors and editors should check whether showing even adapted and cited figures previously published is ok.

Response: We changed the figure 1. Consequently, we changed the description in Figure legend as the underlined sentences.

We’ve obtained the permissions of figures from each journal office.

Reviewer 2 Report

This version has been extensively and adequately revised based on my comments

Author Response

Thank you for your reviewing.
